# The Essential Oil of *Citrus lumia* Risso and Poit. ‘Pyriformis’ Shows Promising Antioxidant, Anti-Inflammatory, and Neuromodulatory Effects

**DOI:** 10.3390/ijms24065534

**Published:** 2023-03-14

**Authors:** Antonella Smeriglio, Susanna Alloisio, Raffaella Barbieri, Mariarosaria Ingegneri, Paola Malaspina, Bruno Burlando, Laura Cornara, Domenico Trombetta

**Affiliations:** 1Department of Chemical, Biological, Pharmaceutical and Environmental Sciences, University of Messina, Viale Ferdinando Stagno d’Alcontres 31, 98166 Messina, Italy; 2ETT Spa, via Sestri 37, 16154 Genova, Italy; 3Institute of Biophysics (IBF), Consiglio Nazionale delle Ricerche (CNR), via De Marini 6, 16149 Genova, Italy; 4Department of Earth, Environment and Life Sciences (DISTAV), University of Genova, Corso Europa 26, 16132 Genova, Italy; 5Department of Pharmacy (DIFAR), University of Genova, Viale Benedetto XV 3, 16132 Genova, Italy

**Keywords:** acetylcholinesterase inhibitory activity, anti-inflammatory activity, antioxidant activity, essential oil, macro and micro-morphological analyses, multi-electrode arrays, neuroinhibitory properties, phytochemical analyses

## Abstract

*Citrus lumia* Risso and Poit. ‘Pyriformis’ are horticultural varieties of *Citrus lumia* Risso. The fruit is very fragrant and pear-shaped, with a bitter juice, a floral flavor, and a very thick rind. The flavedo shows enlarged (0.74 × 1.16 mm), spherical and ellipsoidal secretory cavities containing the essential oil (EO), visible using light microscopy, and more evident using scanning electron microscopy. The GC-FID and GC-MS analyses of the EO showed a phytochemical profile characterized by the predominance of D-limonene (93.67%). The EO showed interesting antioxidant and anti-inflammatory activities (IC_50_ 0.07–2.06 mg/mL), as evaluated by the in vitro cell-free enzymatic and non-enzymatic assays. To evaluate the effect on the neuronal functional activity, the embryonic cortical neuronal networks grown on multi-electrode array chips were exposed to non-cytotoxic concentrations of the EO (5–200 µg/mL). The spontaneous neuronal activity was recorded and the mean firing rate, mean burst rate, percentage of spikes in a burst, mean burst durations and inter-spike intervals within a burst parameter were calculated. The EO induced strong and concentration-dependent neuroinhibitory effects, with IC_50_ ranging between 11.4–31.1 µg/mL. Furthermore, it showed an acetylcholinesterase inhibitory activity (IC_50_ 0.19 mg/mL), which is promising for controlling some of the key symptoms of neurodegenerative diseases such as memory and cognitive concerns.

## 1. Introduction

Plant essential oils (EOs) have flavor, fragrance, and biological properties, making them an important source material for the pharmaceutical, food, and cosmetic industries [1]. The biological role of EOs resides in enhancing a plant’s ecological competition through reproductive organ attractiveness to pollinators and more efficient seed dispersion, and in acting as a defense system against abiotic (UV, temperature, etc.) and biological agents (herbivores, pathogen insects, microbes) [2].

EOs show a high variability among plant species, subspecies, and varieties, which can be exploited through chemotaxonomic studies and as promising method for assessing the genetic diversity of species, quantifying the relationships between the cultivars or species, and classifying unknown cultivars based on the discriminating compounds [3]. The EO variability depends on both endogenous factors, such as the genetic variability, plant organs, and fruit maturity stage [4], and exogenous factors, such as the place of growing and cultivation, plant diseases, and cultural practices [5]. The EO composition is, consequently, highly specific to each taxon, with different proportions of ubiquitarian compounds and the presence of specific constituents [3,6].

EOs have been used in folk medicines since ancient times [7,8]. Due to their high volatility, they are prevalently used in aromatherapy for a series of neurological disorders, including pain, cognitive processes, sleep, anxiety, relaxation, and sedation [9,10]. However, preclinical studies have also shown their therapeutic potential for the treatment of cardiovascular diseases, diabetes, and cancer [2,11]. Their antimicrobial properties have been validated in several studies [12,13], including the synergistic effects of the different EO constituents against human pathogens [14], overall representing an important element for possible medicinal applications.

Fruits from plants belonging to the *Citrus* genus are preferential EO sources rich in bioactive constituents. These fruits have been introduced to the Mediterranean area from Asia for centuries, and spontaneous crosses have generated several hybrid forms, some of which have been agriculturally exploited, while others retained an importance mainly as ornamental products [15]. Many studies on *Citrus* and *Citrus* relatives have been focused on their high taxonomic and genetic variability [16,17,18,19], and the conservation of these taxa plays an important role in maintaining biodiversity in relation to their historical and agronomic profile [20]. However, only recently studies have been carried out on the morphological, anatomical, and phytochemical characteristics of ancient *Citrus* fruits, as well as on the biological properties of their EOs [15,21,22,23,24]. These products are a rich source of bioactive compounds with antioxidant, antimicrobial, anti-inflammatory, and cytoprotective activities [25,26,27]. Moreover, different studies have reported the effects of the EOs from different *Citrus* species on the central nervous system, including the depressant and anticonvulsant effects of *C. limon* in mice [28], the anxiolytic and sedative effects of *C. sinensis* [29], and the antinociceptive effects of *C. limon* in rats [30].

In a previous study on the Southern Italy *Citrus lumia* Risso and Poit., we analyzed the chemical composition of the EO and showed its antioxidant, anti-cholinesterase, and neuroactive properties, suggesting a possible preventive use for oxidative stress-related diseases [23]. However, EOs are complex terpene mixtures known for their compositional variability due to environmental factors and taxonomical differences even between closely related species and varieties [31]. Therefore, in the present study, we focused on the ‘Pyriformis’ variety of the above *C. lumia* (*Citrus lumia* Risso and Poit. ‘Pyriformis’) [32], to evaluate its micro-morphological, phytochemical, and biological activities for the first time and to compare them with those of the parent species. According to what was previously reported [23], after a micromorphological evaluation of the *C. lumia* fruit, the EO was isolated using hydrodistillation and characterized through phytochemical analyses. The antioxidant, anti-inflammatory, and anti-acetylcholinesterase properties were evaluated through in vitro cell-free assays. In addition, the cytotoxic and neuroactive effects were evaluated using the primary cultures of rat embryo cortical neurons on multi-electrode array (MEA) devices.

## 2. Results

### 2.1. Macro- and Micro-Morphological Analyses

At maturity, the fruits appeared pyriform in shape and bright yellow in color (Figure 1A, B), and showed a variability in the dimensions ranging from 8 × 5.5 to 11.5 × 10 cm. The fruit peel consisted of two regions, visible in the transversal section. The epicarp or flavedo was the thin yellow pigmented upper zone, while the mesocarp or albedo was the white layer with approx. a 1–1.3 cm thickness (Figure 1C). In the flavedo, many oil cavities were visible, showing the spherical and ellipsoidal/pyriform shapes (Figure 1D). The variability in the oil cavities was confirmed using scanning electron microscopy (SEM) (Figure 2A–C). The two kinds of oil cavities (namely spherical and pyriform) also showed a variability in their dimensions. Therefore, we measured them at the maturity stage when the enlarged cavity was completely filled with EO. For each type, the dimensions of the polar and equatorial axes were calculated (*n* = 30), as reported in Table 1.

During the fruit development, the specialized layers of the epithelial cells lining the secretory cavities underwent a schizolysigenous process, giving rise to an enlarged cavity where the EO was accumulated (Figure 2D). The EO present within these oil cavities showed a lipophilic nature, as highlighted by the orange-red drops stained with Sudan III (Figure 2E), and by the brilliant yellow fluorescence of the drops with Fluorol Yellow 088, revealing the presence of lipophilic molecules, such as terpenes (Figure 2F). The observation using polarized light showed the presence of many small prismatic calcium oxalate crystals in the epidermal layer of the flavedo (Figure 3A,B). The chemical composition of these crystals was confirmed by the SEM coupled with the energy dispersive spectroscopy (SEM-EDS) analysis (Figure 3C).

### 2.2. Phytochemical Characterization

*C. lumia* ‘Pyriformis’ had a percentage yield of the EO equal to 3.85% (*v*/*w*) and showed an interesting phytochemical profile. The EO composition, together with the Kovats retention index (KI) and the compound percentages (%), are shown in Table 2.

Thirty-four compounds belonging to three main classes were identified. The most abundant compounds were reported in bold (Table 2). Monoterpenes represented the most abundant class (96.45%), followed by the oxygenated monoterpenes (1.11%), sesquiterpenes (0.39%), and other compounds (2.06%) belonging to minor classes. The main constituent was D-limonene, which alone represented 93.80% of the entire EO complex mixture, followed by β-phellandrene (1.55%), β-linalool (0.85%), the most abundant oxygenated monoterpene, β-ocimene (0.74%), and nootkatone (0.66%).

### 2.3. Biological Properties

#### 2.3.1. In Vitro Cell-Free Assays

The antioxidant, anti-inflammatory, and acetylcholinesterase activity of the EO were evaluated using several in vitro cell-free enzymatic and non-enzymatic tests, based on different environments and reaction mechanisms. Table 3 summarizes these results, expressed as a half-maximal inhibitory concentration against the radical, inflammatory, or enzymatic activity (IC_50_), with the relative 95% confidence limits (C.L.). The results in Table 3 are compared to the relative reference standards.

The EO showed a concentration-dependent antioxidant and free radical scavenging activity in four of the six tests, showing the following order of its potency, expressed as IC_50_: ORAC > iron-chelating activity > β-carotene bleaching > FRAP (Table 3). On the contrary, it did not show any activity in the DPPH and TEAC assays within the tested concentration range (Table 3). Moreover, the EO showed a potential anti-inflammatory activity, which inhibited both the dose-dependent heat-induced BSA denaturation and the protease activity (Table 3). Finally, the data showed a moderate, concentration-dependent, inhibitory activity against acetylcholinesterase, with an IC_50_ equal to 0.19 mg/mL. Therefore, this EO deserves further investigation from a neuromodulatory point of view.

#### 2.3.2. Effect on Neuronal Function

To test whether the application of the EO could induce cytotoxic effects, we firstly investigated the cell viability in the cultures of the cortical neurons. The treatment using increasing EO concentrations (5–200 μg/mL) did not show any effect in the cell viability assay (Figure 4F).

After this, the neural networks were exposed to the EO in concentrations of 5, 10, 30, 50, 100, and 200 µg/mL to investigate the potential effects on the spontaneous electrical activity developed by the neuronal networks grown on the MEA chips. The different concentrations were administrated in a cumulative way, starting from the least to the highest concentration, and left each for 20 min. The treatments induced a concentration-dependent inhibition of the functional activity of the neuronal networks (see Appendix A). This was visible from the concentration–response curves at the following parameters: the mean firing rate (MFR), mean burst rate (MBR), percentage of spikes in a burst (% Spikes_B), mean burst duration (MBD; s), and inter-spike intervals within a burst (MISI_B; s) (Figure 4A–E). By fitting each curve, it was possible to obtain the relative IC_50_ values as follows: 22.2 ± 1.9 µg/mL (MFR), 19.5 ± 3.1 µg/mL (MBR), 11.4 ± 1.9 µg/mL (% Spikes_B), 24.5 ± 0.9 µg/mL (MBD), and 31.1 ± 1.5 µg/mL (MISI_B).

## 3. Discussion

*C. lumia* ‘Pyriformis’, known by the common Italian name of ‘Pera del Commendatore’ (Commander’s Pear), shows a typical fruit shape—narrower at one end, and therefore resembling a pear—and characterized by a thick skin that varies from greenish yellow to bright yellow in color [32]. On the other hand, the ancient Mediterranean species, *Citrus lumia* Risso, studied in our previous research, showed an ovoid, mamillate fruit with a very pronounced umbo [23,24,33].

Regarding the secretory cavities, the light microscope and SEM observations confirmed that in *C*. *lumia* ‘Pyriformis’, similar to what was observed in *C. lumia*, the epithelial cells lining the cavities underwent a schizolysigenous process, giving rise to an enlarged cavity where the EO was released (Figure 2). However, in the variety ‘Pyriformis’, there was a greater variability in the cavity shapes at the maturity stage that, when filled with the EO, appeared either spherical and smaller or pyriform and larger (Figure 2).

The EO extraction yield obtained in the present study (3.85%) fell perfectly within the typical range of the *Citrus* genus (0.5–5%) [34]. The quality and quantity of the EO depends on many factors, such as the type of fruit, its origin, genotype, pedo-climatic conditions, and the EO isolation methods [34]. However, if the isolation is carried out immediately after harvesting, the extraction yield considerably increases, avoiding the formation of peroxidation, isomerization, or rearrangement products [35]. Moreover, from the results obtained in the present study and those previously obtained on *C. lumia* Risso [23], a direct correlation between the extraction yield of the EO and the size of the schizolysigenous pockets in which it is contained was hypothesized.

From a phytochemical point of view, the EO of *C. lumia* ‘Pyriformis’ showed a characteristic profile since D-limonene reached the topmost range of the *Citrus* genus, in which this compound is generally the most abundant. Although limonene can be extracted from more than 300 EOs, the *Citrus* genus, with its numerous species, represents the richest source of this precious monoterpene. The content of D-limonene ranges between 68–98% in sweet and bitter oranges, 45–76% in lemons, 52–69% in mandarins, and 32–45% in bergamot EOs [36,37]. Furthermore, comparing the results of the present study to those previously obtained on *C. lumia* Risso, it was possible to observe a substantial variation in terms of the D-limonene content in the EO from the parent species to the horticultural variety (48.91% vs. 93.80%, respectively) [23].

Limonene is a colorless non-oxygenated cyclic monoterpene consisting of two isoprene units and is well known for its pleasant lemon scent and low toxicity (oral LD_50_ values 5–6 g/kg). This compound was not found to induce any mutagenic, carcinogenic, or nephrotoxic risk to humans [38]. It was registered as generally recognized as safe (GRAS) in the Code of Federal Regulations for its use as a synthetic flavoring agent and it is commonly used in several pharmaceutical, food, and cosmetic products, in addition to ecofriendly pesticides and insect repellents [39,40]. In fresh *Citrus* fruits, limonene is generally present in the D-isoform, its main active form [41]. This isoform has several health and therapeutic properties, such as antioxidant, anti-inflammatory, anticancer, anti-asthmatic, anti-microbial, and anticholesterolemic activities, as well as neuroactive and neuroprotective properties [37,39]. This wide range of biological activities is probably due to its marked lipophilicity, which allows it to easily cross all biological membranes, including the intestinal and blood–brain barrier, through simple passive diffusion, providing a good bioavailability in the systemic circulation [41].

Regarding the oxygenated monoterpenes, despite the low concentration of β-linalool (0.83%) found in this horticultural variety, this concentration reflected the range of concentrations generally found in other fruits belonging to the *Citrus* genus (0.02–10.23%) [42]. Furthermore, reviewing the available literature, it appeared that the EO of *C. lumia* Risso had the highest concentration of β-linalool (18.25%) [23].

It is well known that oxidative stress and inflammation are trigger events for many chronic degenerative diseases [43]. Therefore, antioxidants and anti-inflammatory compounds play a pivotal role in preserving human health by restoring the physiological oxidative and inflammatory balance through the modulation of several biological pathways and membrane functions [44,45]. Comparing the results obtained in the present study to those in our previous study on the *C. lumia* Risso EO [23], it can be deduced that, although the two EOs showed different orders of antioxidant potency, both possessed a strong ability to neutralize various radical species, marking them as important sources of antioxidants that could be potentially useful in the detoxification process, especially for the hydrogen-atoms transfer mechanism. Furthermore, as shown by the strong activity found in the β-carotene bleaching test, the antioxidant compounds present in both EOs formed adducts with peroxyl radicals, thus demonstrating a predisposition to hinder the lipid peroxidation in biological membranes. This activity also contributed a strong iron-chelating activity, which reduced the availability of the transition metals and inhibited the Fenton-like oxidative chain reactions in the biological systems, thus preserving the integrity and functionality of the membranes. These biological properties may be attributed to the preponderance of the monoterpenes in both the investigated EOs, since it is well known that these compounds have a greater antioxidant activity than the oxygenated monoterpenes and sesquiterpenes [46]. Considering the anti-inflammatory activity, a direct comparison with the other *Citrus* EOs could not be conducted as the EO of *C. lumia* ‘Pyriformis’ was the only one that was investigated by the two tests performed in the present study. However, through various different methods, the anti-inflammatory activity of *Citrus* EOs has been widely investigated [40].

Several aromatic plants have been investigated to alleviate and ameliorate neuronal disorders. From this point of view, EOs could play an important role in adjuvant therapy, mainly for the pathologies involving the cholinergic system. A strong inhibitory activity on acetylcholinesterase was previously found for the *C. lumia* Risso EO, approx. three times higher than the lemon EO [47] and quite similar to the other *Citrus* species, such as *C. aurantifolia* Swingle, *C. aurantium* L., and *C. bergamia* Risso and Poit. [27]. Interestingly, the EO of *C. lumia* ‘Pyriformis’ showed a stronger inhibition activity that seemed to be directly related to the higher content of the monoterpenes with respect to the parent species. Based on the results of several studies, it was demonstrated that these compounds play a pivotal role in the acetylcholinesterase inhibition [27]. This biological property is related to the hydrophobicity of the monoterpenes, since their cyclic or acyclic hydrocarbon skeleton may interact with the active site of the enzyme [27]. However, it is interesting to note that an IC_50_ value of 586.3 µg/mL was reported for the acetylcholinesterase activity of D-limonene [47], which was almost three times greater than that recorded in the present study, although this was the first compound in the order of abundance (93.80%) in the *C. lumia* ‘Pyriformis’ EO. Furthermore, linalool, the second most abundant compound of the investigated EO, inhibited the acetylcholinesterase activity by 27% at 164 µg/mL [47]. Therefore, these results indirectly confirmed that, although the inhibitory activity on acetylcholinesterase may be mostly attributable to the monoterpenes, the additive and/or synergistic interactions among the different components of an EO play a key role in conferring the final biological activity to the plant complex [23].

EOs are volatile plant complexes composed by hydrophobic substances that diffuse rapidly across biological membranes and easily cross the blood–brain barrier. Therefore, it is particularly interesting to investigate the possible modulatory properties of EOs on the various neuronal functions [48]. Our experiments with cortical neural networks on MEA chips revealed strong inhibitory activities of the *C. lumia* ‘Pyriformis’ EO on all the recorded electrophysiological parameters, without a reduced neuronal vitality. Interestingly, these activities showed a lower IC_50_, i.e., a stronger inhibition, than those observed in our previous study on the *C. lumia* Risso EO [23]. Such a difference could be explained by the much higher content of D-limonene in the *C. lumia* ‘Pyriformis’ EO (about 94%) with respect to the *C. lumia* Risso EO (about 49%). This hypothesis is strongly supported by several studies indicating the neuroprotective properties of the lemon EO, and especially of limonene on different neurological diseases, including Alzheimer’s disease, epilepsy, stroke, multiple sclerosis, and anxiety [37,49,50]. More specifically, the studies on the mechanisms of the action of limonene have shown anti-anxiety and anti-seizure effects through the regulation of dopaminergic and GABAergic functions in mice [51,52]. Moreover, spatial memory improvement, reduced anxiety, and reduced drug addiction have been observed through the regulation of postsynaptic dopamine receptor super-sensitivity in rats [53,54]. Finally, limonene has been found to counteract the ROS production induced by the Aβ oligomers in the rat primary cortical neurons [55]. Therefore, our data, combined with the previous literature reports, provide a strong indication for the possible use of the *C. lumia* ‘Pyriformis’ EO as an alternative agent or an adjuvant to therapeutic drugs against the onset or progression of neuroinflammatory and neurodegenerative diseases, epilepsy, and mood disorders.

## 4. Materials and Methods

### 4.1. Chemicals

Chemicals were purchased from the following suppliers, unless otherwise indicated. Ethanol, histochemical stains (Fluorol Yellow 088; Sudan III), 2,2-diphenyl-1-picrylhydrazyl (DPPH), potassium peroxydisulfate (K_2_S_2_O_8_), 2,2′-azino-bis (3-ethylbenzothiazoline-6-sulfonic acid) diammonium salt (ABTS), 2-4-6-tris(2-pyridyl)-s-triazine (TPTZ), ethylen diaminetetracetic acid (EDTA), iron sulphate heptahydrate (FeSO_4_·7H_2_O), ferrozine, sodium phosphate dibasic (Na_2_HPO_4_), potassium phosphate monobasic (KH_2_PO_4_), sodium acetate (CH_3_COONa), iron(III) chloride hexahydrate (FeCl_3_·6H_2_O), iron(II) chloride tetrahydrate (FeCl_2_·4H_2_O), β-carotene, linoleic acid, Tween 40, chloroform, butylated hydroxytoluene (BHT), 2,2′-azobis(2-methylpropionamidine) dihydrochloride (AAPH), fluorescein disodium salt, C7-C40 saturated alkane standard, acetylthiocholine iodide (ATCI), 5,5′-dithiobis(2-nitrobenzoic acid) (DTNB), galantamine, bovine serum albumin (BSA) heatshock fraction protease, fatty acid and essentially globulin free (pH 7, ≥98%), trypsin from porcine pancreas Type IX-S lyophilized powder (13,000–20,000 BAEE units/mg protein), perchloric acid, trizma-base, casein, diclofenac sodium, and GC-grade dichloromethane were purchased from Merck (Darmstadt, Germany). FineFIX working solution was obtained from Milestone SRL (Sorisole, Bergamo, Italy). The reference standards used for the chemical characterization of the EO shown in Table 1 have been purchased from Extrasynthese (Genay, France). All the chemicals and solvents were of an analytical grade.

### 4.2. Plant Materials

Trees of *C. lumia* ‘Pyriformis’ of horticultural origin have been present at the Hanbury Botanical Gardens of La Mortola, Ventimiglia (Italy) since 2003. Mature fruits (Figure 1A) were collected during November 2019 and immediately sent to the laboratories to carry out macro- and micro-morphological analyses and to isolate the EO.

### 4.3. Macro- and Micro-Morphological Analyses

#### 4.3.1. Light Microscopy (LM)

The mature fruits of *C. lumia* ‘Pyriformis’ were sliced with a knife along the maximum diameter and the transversal section of the exocarp (corresponding to the fruit peel) was made using a razor blade. The macro-morphological details of the peel were firstly highlighted using a stereomicroscope (LEICA M205 C—Leica Microsystems, Wetzlar, Germany). Afterwards, fresh transversal sections were mounted in distilled water and observed under transmission light microscopy to carry out a morphometric analysis of the oil cavities. For this purpose, the polar and equatorial axes of the cavities were measured by using the ToupView software (version x64, 4.11.20805.20220506, ToupTek Photonics, Hangzhou, China). Histochemical assays were performed to analyze the nature of the secretions within the oil cavities. The total lipids were detected using Sudan III, and the presence of the lipids, particularly terpenes, using Fluorol Yellow 088 [56]. In addition, polarized light was used to confirm the presence/absence of crystals in the peel region. Observations were made using a Leica DM 2000 fluorescence microscope equipped with an H3 filter (excitation filter BP 420–490 nm) and a ToupCam Digital Camera, CMOS Sensor 3.1 MP resolution (ToupTek).

#### 4.3.2. Scanning Electron Microscopy (SEM)

Small pieces of fresh mature fruits, measuring about 1 cm^2^, were fixed in a 70% ethanol-FineFix solution for 24 h at 4 °C and dehydrated through a series of increasing ethanol solutions. The samples were then critically point dried (K850-CPD 2M, Strumenti S.r.l., Roma, Italy), mounted on stubs using two-sided adhesive carbon tape, and sputtered with a 10-nm layer of gold. Observations were carried out using a SEM VEGA3-Tescan-type LMU (Tescan USA Inc., Cranberry Twp, PA, USA) operating at an accelerating voltage of 20 kV. Energy dispersive spectroscopy (EDS) coupled with SEM was used to obtain the elemental composition of the crystals.

### 4.4. Isolation of Essential Oil

Fresh mature fruits were manually peeled and the flavedo were subjected to hydrodistillation using a Clevenger-type apparatus, according to the European Pharmacopoeia [57], until no significant increase in the volume of the collected EO was observed (3 h). The extraction yield was 3.85% (*v*/*w*). The EO was dried overnight on anhydrous Na_2_SO_4_ and stored at −20 °C in a dark sealed vial with a N_2_ headspace until the analysis. One hundred milligrams of the EO were dissolved into 1 mL of DMSO to obtain a stock solution (100 mg/mL) that was properly diluted in the cell culture medium (DMSO ≤ 0.01) for the cell-based assays, and in methanol to carry out the in vitro cell-free assays.

### 4.5. Phytochemical Analyses

Phytochemical characterization was carried out using a gas chromatograph model 7890A (Agilent Technologies, Santa Clara, CA, USA) coupled with flame ionization and mass spectrum detectors (GC-FID and GC-MS analysis, respectively). Separation was carried out on a Zebron™ ZB-5MS GC Column 30 m × 0.25 mm × 0.25 µm (Phenomenex, Bologna, Italy) using helium as the carrier gas (0.7 mL/min). Injection (1 µL, 10% EO/CH_2_Cl_2_, *v*/*v*) was performed at 250 °C in split mode (50:1). The oven temperature program was set as follows: 60 °C for 5 min, 100 °C at 3 °C/min, 180 °C at 1 °C/min, 240 °C at 3 °C/min, and held at 240 °C for 5 min.

The detector temperature was set at 280 °C and 180 °C for FID and MS, respectively. The ionization voltage, electron multiplier, and ion source temperature for the MS analysis were set at 70 eV, 900 V, and 230 °C, respectively. Mass spectra were acquired in the full scan mode (m/z 40–450). The detected compounds were identified based on the following parameters: the GC retention index (relative to C7-C40 n-alkanes on Zebron™ ZB-5MS GC Column), matching of mass spectra with those reported in the MS library (NIST 08), comparison of the MS fragmentation patterns with those reported in the literature [58], and, whenever possible, co-injection with the commercially available standards (see Table 1). Quantification was carried out using GC-FID analysis and the results were expressed as the mean peak area percentage ± the standard deviation of three independent experiments in triplicate (*n* = 3).

### 4.6. In Vitro Cell-Free Assays

The EO antioxidant, anti-inflammatory, and acetylcholinesterase activities were evaluated through in vitro colorimetric and enzymatic assays based on the different mechanisms and reaction environments. The results, which represent the average of three independent experiments in triplicate (*n* = 3), were expressed as the inhibition percentage (%) of the oxidative/inflammatory/enzyme activity, calculating the IC_50_ with the corresponding C.L. at 95% from the Litchfield and Wilcoxon’s test using the PHARM/PCS software version 4 (MCS Consulting, Wynnewood, PA, USA). All the concentration ranges reported referred to the final concentrations of the EO and reference compounds in the reaction mixture.

#### 4.6.1. DPPH Assay

The scavenging activity against the DPPH radical was evaluated according to Smeriglio et al. [59] with some modifications. Briefly, 150 µL of the 1 mM DPPH methanol solution was mixed with 3.75 μL of the EO methanol solution (0.25–2 mg/mL), shaken, and left for 20 min at room temperature (RT) in the dark. The absorbance was recorded using an UV/VIS microplate reader (Multiskan GO; Thermo Scientific, Waltham, MA, USA) against a blank consisting of methanol and using trolox as the reference standard (1.25–10 μg/mL).

#### 4.6.2. TEAC Assay

The TEAC assay was carried out according to Danna et al. [60] with some modifications. Briefly, the reagent consisting of 1.7 mM of ABTS and 4.3 mM of ammonium persulfate (1:5, *v*/*v*) was incubated in the dark at RT for 12–16 hrs, diluted with deionized water to obtain an absorbance ranging from 0.680 and 0.720 at 734 nm, and then used within 4 hrs. Ten microliters of the EO methanol solution (0.30–2.4 mg/mL) were added to 200 μL of the above reagent and left for 6 min at RT in the dark. The absorbance was recorded by the same instrument and the same blank reported in the Section 4.6.1. Trolox was used as reference standard (0.625–5.0 μg/mL).

#### 4.6.3. FRAP Assay

The ferric ion reducing antioxidant power test was carried out according to Denaro et al. [61] with some modifications. Briefly, 10 μL of the EO methanol solution (0.3–2.4 mg/mL) was added to 200 μL of the fresh pre-warmed (37 °C) reagent consisting of a 300 mM buffer acetate (pH 3.6), 10 mM TPTZ-40 mM of HCl and 20 mM of FeCl_3_ (10:1:1, *v*/*v*/*v*), and incubated for 4 min. The absorbance was recorded at 593 nm using the same instrument and blank reported in Section 4.6.1. Trolox was used as the reference standard (0.625–5.0 μg/mL).

#### 4.6.4. Iron-Chelating Activity

The iron-chelating capacity was evaluated according to Smeriglio et al. [62]. Briefly, 25 μL of 2 mM FeCl_2_·4H_2_O were added to 50 μL of the EO methanol solution (0.1–0.8 mg/mL) and incubated at RT for 5 min. Then, 50 μL of a 5 mM ferrozine solution was added and the mixture volume was brought to 1.5 mL using deionized water. After 10 min of incubation at RT, the absorbance was recorded at 562 nm by an UV–VIS spectrophotometer (Shimadzu UV-1601, Kyoto, Japan) using the EDTA as the reference standard (1.25–10.0 μg/mL).

#### 4.6.5. β-Carotene Bleaching

The β-carotene bleaching assay was performed using an emulsion prepared according to Muscarà et al. [63]. Briefly, 8 mL of the emulsion was dispensed into borosilicate glass test tubes to which 0.32 mL of the EO methanol solution (0.25–2.0 mg/mL) was added. The absorbance of the reaction mixture was recorded at 470 nm using the same instrument reported in Section 4.6.4. at the starting time and after 120 min of incubation at 50 °C in a water bath, recording the absorbance every 20 min. The BHT was used as the reference standard (30 μg/mL). A positive control (emulsion with β-carotene but without a sample/standard) and a negative control (emulsion without β-carotene and without a sample/standard) were used to rule out any possible interference.

#### 4.6.6. ORAC Assay

The ORAC test was carried out according to Bazzicalupo et al. [64]. Briefly, 20 μL of the EO solution (0.01–0.08 mg/mL) and trolox (0.25–2.50 μg/mL) as the reference standard, both diluted in a 75 mM phosphate buffer (pH 7.4), were mixed with 120 μL of a fresh 117 nM fluorescein solution and incubated in the dark at 37 °C for 15 min. Then 60 μL of a fresh 40 mM AAPH solution was added, triggering the radical reaction. The fluorescence was recorded using a microplate reader (Fluostar Omega, BMG Labtech, Ortenberg, Germany) every 30 s for 90 min at the following excitation and emission wavelength: λ_ex_ = 485 nm and λ_em_ = 520 nm, respectively.

#### 4.6.7. Bovine Serum Albumin (BSA) Denaturation Assay

The ability of the EO to inhibit the heat-induced BSA denaturation was evaluated according to Smeriglio et al. [65]. Briefly, 100 μL of a 0.4% BSA solution and 20 μL of a phosphate buffered saline (PBS, pH 5.3) were added into a 96-well plate. Then, 80 μL of the EO methanol solution (0.031–0.25 mg/mL) and diclofenac sodium as the reference standard (3.0–24.0 μg/mL) were added to the mixture. The absorbance was recorded at 595 nm, immediately and after 30 min incubation at 70 °C, by using the same instrument and blank reported in Section 4.6.1.

#### 4.6.8. Protease Inhibition Assay

The protease inhibitory activity was evaluated according to Smeriglio et al. [33]. Briefly, 12 μL of trypsin (10 μg/mL) and 188 μL of a 25 mM Tris-HCl buffer (pH 7.5) were added to 200 μL of the EO methanol solution (0.031–0.25 mg/mL) and diclofenac sodium reference standard (2–16 μg/mL) and incubated for 5 min at 37 °C in a water bath. Two hundred microliters of 0.8% casein were added, and after 20 min of incubation at 37 °C, 400 μL of 2 M perchloric acid was added as the stopping reagent. The cloudy suspension was centrifuged at 10,000× *g* for 10 min and the absorbance of the supernatant was recorded at 280 nm using the same instrument and blank reported in Section 4.6.4.

#### 4.6.9. Acetylcholinesterase Inhibition Assay

The acetylcholinesterase inhibitory activity was evaluated according to Smeriglio et al. [23]. Briefly, 100 µL of the EO methanol solution (0.05–0.40 µg/mL) and galantamine reference standard (1–8 µg/mL) were dispensed into borosilicate glass test tubes in which 400 µL of PBS (pH 8), 100 µL of 0.4 U/mL acetylcholinesterase, and 200 µL of fresh 0.6 mM DTNB were added. After 10 min of incubation at 37 °C in a water bath, 200 µL of fresh 0.6 mM ATCI was added and incubated again for 30 min. At the end of incubation time, 500 µL of anhydrous ethanol was added as the stopping reagent. The absorbance was recorded at 412 nm using the same instrument and blank reported in Section 4.6.4.

### 4.7. In Vitro Cell-Based Assays

#### 4.7.1. Primary Neuronal Cultures

The neuronal cultures were obtained from the cerebral cortices of the fetal day 17 Wistar SPF rats of either sex, according to previously described procedures [66]. The cortices were mechanically dissociated, using two fire-polished Pasteur pipettes of decreasing diameter, in 5 mL of Hank’s Balanced Salt Solution without Ca^2+^ and Mg^2+^ (Thermo Fisher Scientific, Waltham, MA, USA), and the cell homogenate was allowed to settle. Subsequently, the supernatant was discarded, and the pellet was gently resuspended in a fresh neurobasal medium (NB; Thermo Fisher Scientific) supplemented with 2% B27 (Thermo Fisher Scientific) and 1% glutamine (Sigma Aldrich, Milan, Italy). The cell suspension was opportunely diluted with fresh neurobasal medium and settled on 96 multi-well plates for the viability assay or on MEA chips for the electrophysiology measurements.

#### 4.7.2. Cell Viability Assay

The neuronal cell suspensions were seeded into each well of the 0.1% polyethyleneimine (PEI) pre-coated 96-well plates (3 × 10^3^ cells per well) and maintained in the neurobasal medium (NB) supplemented with 2% B27 and 1% Glutamax-I in a humidified incubator at 37 °C and with a 5% CO_2_ enriched atmosphere. A half volume of the medium was substituted three times a week. After 21 days, the cells were exposed to the EO (5–200 μg/mL) for 2 h. Thereafter, a 20 μL volume of a 3-(4,5-dimethylthiazol-2-yl)-2,5-diphenyltetrazolium bromide solution (MTT, Sigma-Aldrich) was added to each well. The cells were incubated for 2 h at 37 °C, 5% CO_2_ then treated with the MTT solvent (DMSO), and the absorbance was measured at 570 nm. The cell viability was expressed as the percentage of the viable cells compared to the untreated cells.

#### 4.7.3. Electrophysiological Data Recordings, Signal Processing, and Data Analysis

A neuronal cell suspension containing 50,000–60,000 cells was left to settle for 1 h on the center of the 0.1% polyethyleneimine (PEI) pre-coated, 60-electrode MEA chips with internal reference (60MEA200/30iR-Ti-gr; Multi Channel Systems, MCS GmbH, Reutlingen, Germany). Thereafter, 1000 µL of the pre-warmed neurobasal medium (NB) supplemented with 2% B27 and 1% Glutamax-I medium were added to each MEA chip. The cultures were maintained in a humidified incubator at 37 °C with a 5% CO_2_ enriched atmosphere, and a half volume of the medium was substituted three times a week. The cells were maintained in vitro from 4 to 6 weeks before the experiments were carried out.

The electrophysiological measurements were started by placing the MEA chips into the MEA Amplifier (Gain 1000×) of the USB MEA 120 INV 2 BC System (MCS, Reutlingen, Germany), and the cells on the MEA were exposed to the EO in the range 5–200 μg/mL. The data were recorded by the MC_Rack software (MCS, Version 4.4.1.0) at a sampling rate of 10 kHz. A band pass digital filter (60–4000 Hz) was applied to the raw signal to remove electrical background noise. Only the electric signal (spontaneous electrical activity) above the spike detection threshold (i.e., 5.5 times the standard deviation of the mean square root noise) was identified and recorded. The system also included a temperature controller (TC02, MCS GmbH) that maintained the cell culture at 37 °C during the experimentation.

The analysis was conducted using the NeuroExplorer software version 4.135 (Nex Technologies, Colorado Springs, CO, USA) and considered the following parameters: MFR (number of spikes/s), MBR (number of bursts/min), % Spikes_B, MBD (s), and MISI_B (s). The following burst definition parameters were set: the bin size = 1 s; maximum interval of starting a burst = 0.01 s; maximum interval of ending a burst = 0.075 s; minimum burst interval = 0.1 s; minimum burst duration = 0.02 s; minimum number of spikes in a burst = 4. Only the channels with >2 bursts/min were included in the analysis.

To obtain the estimated IC_50_ values, the normalized concentration–response curves of the single treatments were interpolated by a four-parameter logistic function using SigmaPlot 8 (Jandel Scientific):f(x) = Max + (Min − Max)/(1 + (ε/x)^β^)(1)
where the variable x is the concentration of the compound; the parameter Min is the minimum effect; the parameter Max is the maximum effect; the parameter ɛ is the concentration at the inflection point of the concentration–response curve, i.e., the concentration at which the effect is reduced by 50% (IC_50_); and β is the parameter related to the maximum slope of the curve, which occurs at concentration ɛ.

### 4.8. Statistical Analyses

The results were expressed as the mean inhibition percentage, calculating the IC_50_ and the corresponding C.L. at 95% of three independent experiments in triplicate for the in vitro cell-free assays, and as the mean ± the standard error (S.E.) of five independent experiments in triplicate for the cell-based experiments. The statistical significance was evaluated using one-way analysis of variance (ANOVA) followed by a Tukey’s test for the phytochemical and in vitro cell-free analyses, and a two-tailed Dunnett’s t-test for the cell-based assays, using the SigmaPlot 12.0 software. The statistical significance was considered at *p* < 0.05.

## 5. Conclusions

*C. lumia* ‘Pyriformis’ is a horticultural variety of *C. lumia* Risso but its EO shows important phytochemical differences with respect to the parental species, with a significant increase in monoterpenes, especially D-limonene. The EO showed promising antioxidant and anti-inflammatory activities, preventing radical damage and showing interesting iron-chelating and anti-peroxidative properties. However, the most striking results of this horticultural variety with respect to the *Citrus lumia* Risso EO regards the stronger inhibitory effects exerted on the acetylcholinesterase and electrophysiological activities in subtoxic doses. These results are likely due to the pivotal role of D-limonene as a neuroactive substance, suggesting that the *C. lumia* ‘Pyriformis’ EO could represent a turning point for the control and prevention of neurological and neurodegenerative disorders. Moreover, these data highlight the importance of studying old cultivars at risk of extinction due to the loss of agricultural interest. These botanical entities can have peculiar phytochemical profiles making them very interesting sources of medicinal and nutraceutical bioactive compounds.

## Figures and Tables

**Figure 1 ijms-24-05534-f001:**
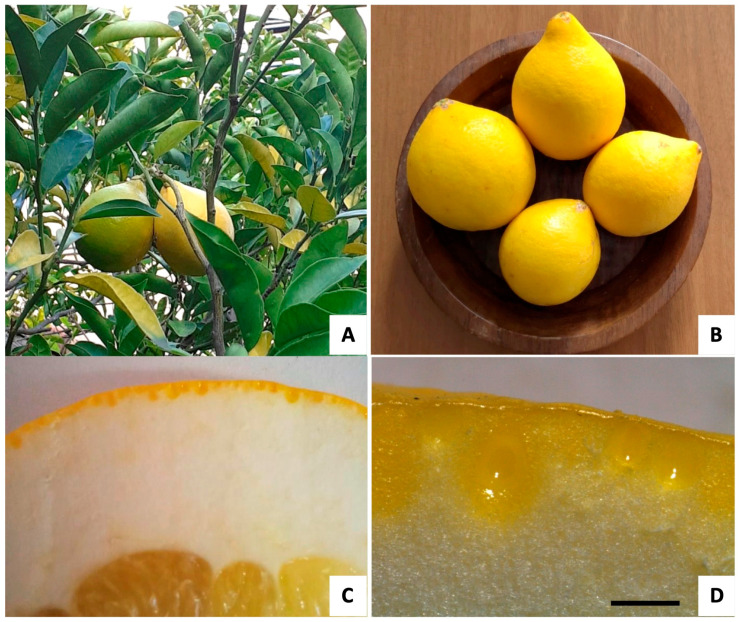
*Citrus lumia* ‘Pyriformis’. (**A**) Plant growing at the Hanbury Botanical Gardens (Ventimiglia, Italy). (**B**) Mature fruits. (**C**) Transversal section of the peel, showing the pigmented flavedo and the white albedo. (**D**) Detail of the flavedo with spherical to ellipsoidal/pyriform oil cavities. Scare bar: 1 mm.

**Figure 2 ijms-24-05534-f002:**
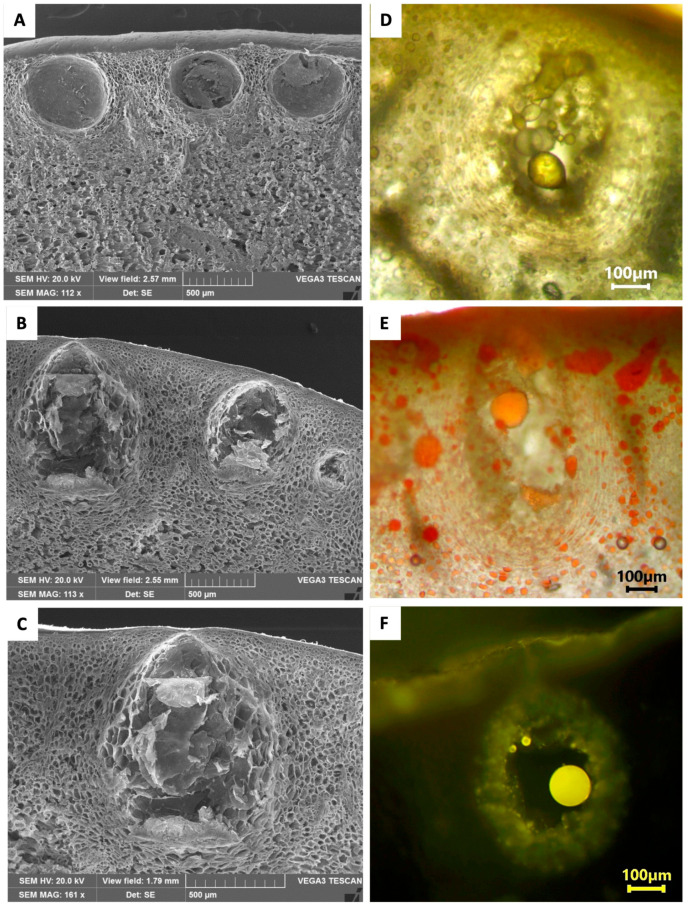
Micrographs of the fruit peel of *C. lumia* ‘Pyriformis’ obtained using SEM (**A**–**C**) and light (**D**–**F**) microscopy. (**A**–**C**): spherical (**A**) to ellipsoidal/pyriform (**B**,**C**) oil cavities visible in a fruit peel cross section. (**D**–**F**): Deposits of the EO visible within the secretory cavities: (**D**) EO droplets without histochemical staining; (**E**) droplets of the EO stained orange-red by Sudan III; (**F**) brilliant yellow fluorescence of the EO drops stained with Fluorol Yellow 088.

**Figure 3 ijms-24-05534-f003:**
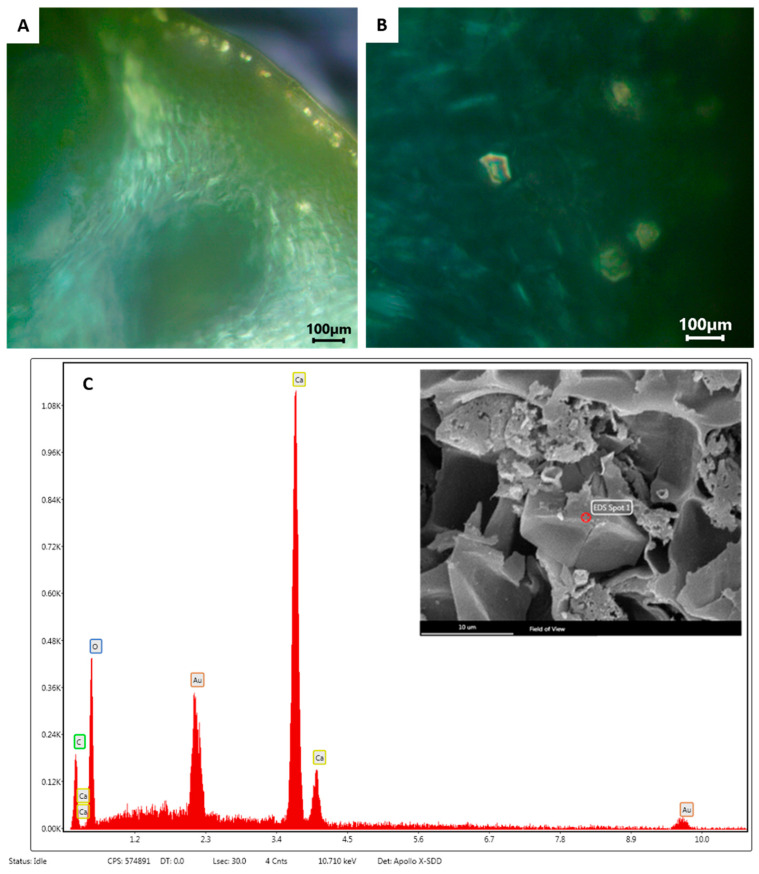
Cross section of *C. lumia* ‘Pyriformis’ peel fruit. (**A**,**B**) Polarized light micrographs showing the presence of many small prismatic calcium oxalate crystals in the epidermal layer of the flavedo. Scare bar: 100 μm (**C**) SEM observation showing a single prismatic crystal in the epidermal layer (inset, scare bar: 10 μm), and corresponding EDS spectrum showing a high calcium level.

**Figure 4 ijms-24-05534-f004:**
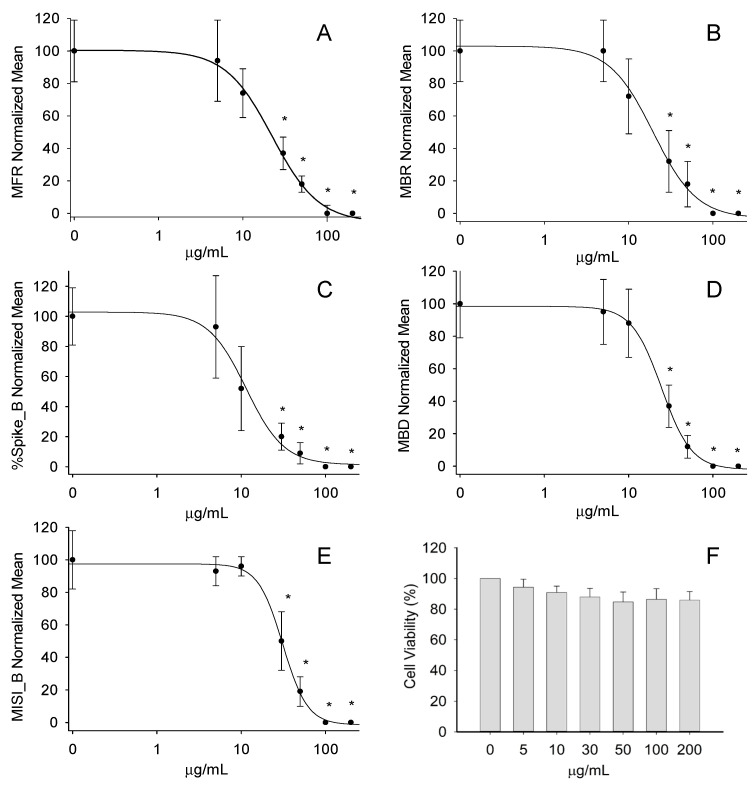
Effect of increasing EO concentrations (5–200 μg/mL) on the neural network functional activity and viability. The EO showed a concentration-dependent inhibition of all the selected parameters describing the neural network electrical activity: (**A**) mean firing rate (MFR), (**B**) mean burst rate (MBR), (**C**) percentage of spikes in a burst (% Spikes_B), (**D**) mean burst duration (MBD; s), (**E**) mean inter-spike interval within a burst (MISI_B; s) (*n* = 6; * *p* < 0.05). At the bottom right (**F**), the bar graph shows that the EO did not induce any significant effect on the neuronal viability evaluated by means of the MTT assay (*n* = 12).

**Table 1 ijms-24-05534-t001:** Dimensions of *C. lumia* ‘Pyriformis’ oil cavities at the maturity stage measured on light micrographs.

Oil Cavity Shapes	Mean Polar Axis	Mean Equatorial Axis
Pyriform/ellipsoidal	1.16 ± 0.15	0.88 ± 0.12
Spherical	0.75 ± 0.2	0.74 ± 0.2

Data are the mean ± SD expressed in mm (*n* = 30).

**Table 2 ijms-24-05534-t002:** Chemical composition of the *C. lumia* ‘Pyriformis’ EO. Results are expressed as the mean area percentage (%) ± the standard deviation (SD) of three independent determinations in triplicate (*n* = 3).

Compound	KI ^a^	Identification ^b^	Area (%)
α-Pinene	939	1, 2, 3	0.28 ± 0.01
β-Pinene	979	1, 2, 3	0.04 ± 0.00
Octanal	998	1, 2	0.01 ± 0.00
α-Phellandrene	1002	1, 2, 3	0.01 ± 0.00
**β-Phellandrene**	1025	1, 2	**1.55 ± 0.03**
**D-Limonene**	1029	1, 2, 3	**93.80 ± 0.24**
**β-Ocimene**	1037	1, 2	**0.74 ± 0.02**
β-Terpinene	1042	1, 2	0.02 ± 0.00
γ-Terpinene	1059	1, 2, 3	0.01 ± 0.00
cis-Linalool oxide	1067	1, 2	0.24 ± 0.01
1,2-Oxolinalool	1085	1, 2	0.11 ± 0.01
**β-Linalool**	1095	1, 2, 3	**0.85 ± 0.02**
Trans-p-Mentha-2,8-dienol	1119	1, 2	0.03 ± 0.00
Limonene oxide trans	1142	1, 2	0.01 ± 0.00
Citronellal	1153	1, 2, 3	0.14 ± 0.01
1-Terpinen-4-ol	1177	1, 2, 3	0.06 ± 0.00
α-Terpineol	1188	1, 2	0.02 ± 0.00
trans-Carveol	1216	1, 2	0.29 ± 0.01
cis-Carveol	1229	1, 2	0.03 ± 0.00
Neral	1238	1, 2	0.11 ± 0.00
(S)-(+)-Carvone	1243	1, 2, 3	0.13 ± 0.01
cis-Geraniol	1252	1, 2, 3	0.03 ± 0.00
Geranial	1267	1, 2	0.04 ± 0.00
Perillal aldehyde	1271	1, 2	0.18 ± 0.01
Indole	1291	1, 2	0.03 ± 0.00
α-Cubebene	1351	1, 2	0.03 ± 0.00
Geraniol acetate	1381	1, 2	0.07 ± 0.00
β-Cubebene	1388	1, 2	0.03 ± 0.00
β-Caryophyllene	1419	1, 2, 3	0.24 ± 0.01
α-Caryophyllene	1454	1, 2, 3	0.02 ± 0.00
Germacrene D	1481	1, 2	0.04 ± 0.00
δ-Cadinene	1523	1, 2	0.04 ± 0.00
trans-Farnesol	1743	1, 2	0.12 ± 0.01
**Nootkatone**	1806	1, 2, 3	**0.66 ± 0.03**
Monoterpene hydrocarbons			96.45
Oxygenated monoterpenes			1.11
Sesquiterpene hydrocarbons			0.39
Others			2.06

^a^ Linear retention index on Zebron™ ZB-5MS GC Column 30 m × 0.25 mm × 0.25 µm (Phenomenex, Bologna, Italy); ^b^ Identification method: 1 = Kovats retention index; 2 = comparison of mass spectra; 3 = co-injection with standard compounds. Major compounds in bold.

**Table 3 ijms-24-05534-t003:** Determination of the antioxidant and anti-inflammatory activities of the *C. lumia* ‘Pyriformis’ EO (CLPEO) using several in vitro assays based on different environment and reaction mechanisms. The results are expressed as a half-maximal inhibitory concentration (IC_50_) with confidence limits at 95% in parentheses, representing the mean of three independent experiments in triplicate, and are compared to the reference standards (RS).

Assay	CLPEO (mg/mL)	RS (µg/mL)
2,2-diphenyl-1-picrylhydrazyl (DPPH)	n.a. ^b^	8.57 (7.32–9.67)
Trolox equivalent antioxidant capacity (TEAC)	n.a. ^b^	3.87 (2.25–4.85)
Ferric reducing antioxidant power (FRAP)	2.06 (1.15–3.44) *	5.39 (4.22–6.67)
Iron-chelating activity (Ferrozine)	0.56 (0.28–0.89) *	10.57 (7.88–12.69)
β-Carotene bleaching (BCB)	1.55 (0.65–2.57) *	0.28 (0.15–0.41)
Oxygen radical absorbance capacity (ORAC)	0.07 (0.04–0.09) *	0.68 (0.35–0.92)
BSA ^a^ denaturation	0.19 (0.17–0.21) *	17.77 (15.23–19.65)
Protease inhibitory activity	1.97 (0.85–3.07) *	8.75 (6.33–10.21)
Acetylcholinesterase inhibitory activity	0.19 (0.17–0.22) *	4.54 (2.89–6.24)

^a^ BSA, Bovine serum albumin. ^b^ n.a., not active. Reference standards: Trolox for the DPPH, TEAC, FRAP, and ORAC assays; butylated hydroxytoluene (BHT) for the β-carotene bleaching assay; ethylenediaminetetraacetic acid (EDTA) for the ferrozine assay; diclofenac sodium for the anti-inflammatory assays; and galantamine for the acetylcholinesterase inhibitory activity test. * = *p* < 0.001 vs. RS.

## Data Availability

Not applicable.

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
