# Peer review of "The Essential Oil of Citrus lumia Risso and Poit. ‘Pyriformis’ Shows Promising Antioxidant, Anti-Inflammatory, and Neuromodulatory Effects"

_ijms, 2023, doi:10.3390/ijms24065534_

Round 1
Reviewer 1 Report
In this manuscript, Smeriglio et al analyzed the major components and potential activities of C. Lumia “Pyriformis” essential oil. Some comments:
1. Where are the results of DPPH and TEAC assays? They are not in Table 2.
2. How much essential oil can be extracted from C. Lumia “Pyriformis”?
3. D-limonene is most abundant compound in the C. Lumia “Pyriformis” essential oil. There should be more comparison and discussion between essential oil and D-limonene.
4. The authors especially highlighted that the C. Lumia “Pyriformis” essential oil could be used to control neurodegenerative diseases. Are there any applications which showed that C. Lumia “Pyriformis” or its major components can help to control neurodegenerative diseases?
Author Response
Authors. We thank the reviewer for her/his accurate revision of the manuscript.
English language and style are fine/minor spell check required X
Reply: The English language has been thoroughly revised
- Where are the results of DPPH and TEAC assays? They are not in Table 2.
Reply: We did not report the results because, as mentioned within the text of the pdf file at line 185-186, the EO is not active in these two assays. However, we would like to thank the Reviewer for his comment that pointed out the necessity to explicit, for clarity, these results in Table 2.
- How much essential oil can be extracted from C. Lumia “Pyriformis”?
Reply: The extraction yield of the EO was already reported in the first version of the manuscript. However, we thank the Reviewer for his suggestion and add the extraction yield also in the materials and methods, section 4.4. in order to render it more visible.
- D-limonene is most abundant compound in the C. Lumia “Pyriformis” essential oil. There should be more comparison and discussion between essential oil and D-limonene.
Reply: The discussion about essential oil and D-limonene was implemented as suggested. New references have been also added.
- The authors especially highlighted that the C. Lumia “Pyriformis” essential oil could be used to control neurodegenerative diseases. Are there any applications which showed that C. Lumia “Pyriformis” or its major components can help to control neurodegenerative diseases?
Reply: A reference about the anti-neurodegenerative property of limonene was already present, but others have been now added. No news is instead available about the anti-neurodegenerative properties of C. lumia pyriformis essential oil, this is a final hypothesis of the present article, supported by the combination of literature reports and our new data, and therefore, it pertains to the originality of the article itself.
Reviewer 2 Report
Interesting observational on the field of molecular plants sciences.
Author Response
Authors: We thank the reviewer for her/his accurate revision of the manuscript.
Introduction can be improved X
Results can be improved X
Reply: New statements and references have been added
Reviewer 3 Report
Smeriglio et al submitted a manuscript titled "The essential oil of Citrus lumia Risso & Poit. 'Pyriformis' modulates the functional activity of rat embryo cortical neural networks" for publication in MDPI IJMS.
While the work is interesting that is more focused on essential oil part, essentially, the results mentioned in the title are reflected only in section 2.3.2. "The treatments induced a concentration-dependent in-201 hibition of the global functional activity of the neuronal networks " doesn't mean anything significant.
The ms doesn't fit in the scope and impact of this journal.
Author Response
Authors. We thank the reviewer for her/his accurate revision of the manuscript.
Moderate English changes required X
Reply: we have thoroughly revised the English language.
Introduction must provide further background and reference X
References must be improved X
Reply: New references have been added to better support our discussions
Results must be improved X
Conclusions supported by results must be improved X
Reply: New statements and references have been added
While the work is interesting that is more focused on essential oil part, essentially, the results mentioned in the title are reflected only in section 2.3.2. "The treatments induced a concentration-dependent inhibition of the global functional activity of the neuronal networks " doesn't mean anything significant.
Reply: The term “global” has been deleted, thus making it clear that the statement refers to the series of electrophysiological parameters measured in the neural network, listed in the following text. The results mentioned in the title actually derive from the combination of our phytochemical and electrophysiological analyses.
The discovery that the essential oil almost totally consists of limonene allowed us to ascribe to this compound the effects observed on the neuronal electrophysiological activity, with a high degree of confidence.
The MS doesn't fit in the scope and impact of this journal.
Reply: The manuscript fits with the topics of the special issue entitled “Phytochemicals from Aromatic and Medicinal Plants: From Identification to Biomedical Applications 2.0”. The manuscript has been evaluated and approved for submission by the Editorial board.
Round 2
Reviewer 1 Report
The authors addressed my comments. no more comment.
Author Response
The authors addressed my comments. no more comment.
Authors. We thank the reviewer for her/his positive comment.
Reviewer 3 Report
The quality of manuscript is not the concern here. The work is majorly on the essential oils, and little to do with its effects on neural network. This work doesn't qualify for IJMS.
Author Response
Rev. The quality of manuscript is not the concern here. The work is majorly on the essential oils, and little to do with its effects on neural network. This work doesn't qualify for IJMS.
Authors. The article has been submitted upon Journal’s invitation to the IJMS SI “Phytochemicals from aromatic and medicinal plants, from identification to biomedical applications 2.0”. Therefore, we believe that the article falls within the topics of the SI. However, we acknowledge that the previous title did not perfectly reflect the article contents, and we have modified it accordingly.